# Molecular Pathogenesis of Myeloproliferative Neoplasms: From Molecular Landscape to Therapeutic Implications

**DOI:** 10.3390/ijms23094573

**Published:** 2022-04-20

**Authors:** Erika Morsia, Elena Torre, Antonella Poloni, Attilio Olivieri, Serena Rupoli

**Affiliations:** Clinica di Ematologia, Ospedali Riuniti di Ancona, Via Conca 71, 60126 Ancona, Italy; torre_elena@libero.it (E.T.); a.poloni@univpm.it (A.P.); a.olivieri@univpm.it (A.O.); serena.rupoli@ospedaliriuniti.marche.it (S.R.)

**Keywords:** myeloproliferative neoplasms, JAK/STAT pathway, additional mutations

## Abstract

Despite distinct clinical entities, the myeloproliferative neoplasms (MPN) share morphological similarities, propensity to thrombotic events and leukemic evolution, and a complex molecular pathogenesis. Well-known driver mutations, *JAK2*, *MPL* and *CALR*, determining constitutive activation of JAK-STAT signaling pathway are the hallmark of MPN pathogenesis. Recent data in MPN patients identified the presence of co-occurrence somatic mutations associated with epigenetic regulation, messenger RNA splicing, transcriptional mechanism, signal transduction, and DNA repair mechanism. The integration of genetic information within clinical setting is already improving patient management in terms of disease monitoring and prognostic information on disease progression. Even the current therapeutic approaches are limited in disease-modifying activity, the expanding insight into the genetic basis of MPN poses novel candidates for targeted therapeutic approaches. This review aims to explore the molecular landscape of MPN, providing a comprehensive overview of the role of drive mutations and additional mutations, their impact on pathogenesis as well as their prognostic value, and how they may have future implications in therapeutic management.

## 1. Introduction

Myeloproliferative neoplasms (MPNs) are an heterogenous group of clonal hematopoietic disorders characterized by myeloid progenitor proliferation in the bone marrow, which involve an excess of differentiated erythrocytes, platelets and leukocytes circulating in peripheral blood [1]. MPNs share many features, including a similar mutational landscape, a propensity to thrombosis and hemorrhage, and a risk of leukemic transformation in the long term. According to WHO 2016 classification, BCR-ABL negative MPNs are classified into different nosological categories as polycythemia vera (PV), essential thrombocythemia (ET), primary myelofibrosis (PMF, overt fibrotic and prefibrotic stage), and other rare disorders as chronic neutrophilic leukemia (CNL), chronic eosinophilic leukemia, not otherwise specified (CEL, NOS) and unclassifiable MPN (MPN-u) [2,3]. The diagnostical criteria of secondary myelofibrosis (post-PV-MF and post-ET-MF) were developed by International Working Group for MPN Research and Treatment [4]. While the prevalence remains difficult to determine, in Europe the incidence of MPN varies from 0.4 to 2.8/100.000 in patients affected by PV, from 0.38 to 1.7/100.000 in ET patients and from 0.1 to 1/100.000 in PMF [5].

The hallmark of MPN is the clonal hematopoiesis driven by acquired somatic mutations in myeloid progenitor cells, in particular phenotypic driver mutations in *JAK2*, *CALR* and *MPL* genes induce constitutive activation of intracellular JAK-STAT pathway. *JAK2* V617F mutation is detected in 95% of patients with PV and it is present in approximately 50% of ET and PMF. *CALR* and *MPL* are mutated in most remaining patients with ET and PMF, while “triple negative” patients make up a small part of ET and PMF cases. Moreover, modern sequencing efforts have identified the complex genomic landscape of MPN with additional genetic alterations, especially in epigenetic modifiers and splicing factors [6]. The updated WHO 2016 criteria emphasize the presence of these genetic aberrations to conform a suspected diagnosis of MPN. 

## 2. Driver Mutations in MPN

The principal and mutually exclusive mutations in MPNs occur in *JAK2*, *CALR* and *MPL* and they converge on Janus kinase (JAK)-signal transducer and activator of transcription (STAT) signaling. Mutations that activate the JAK-STAT signaling pathway are sufficient to cause MPN. JAK-STAT pathway plays several and critical roles in adapting of immune system, especially by modulating the polarization of T helper cells and by expressing cytokine receptors on cell surface. In the canonical pathway, JAKs are activated upon cytokine stimulation, in particular on binding of ligand to type I cytokine receptors including thrombopoietin (TPO) receptor MPL, colony-stimulating factor (G-CSF) receptor, and erythropoietin (EPO) receptors. Activation of JAK determines the phosphorylation of STATs which results in dimerization and translocation of STATs to the nucleus in order to activate or suppress the transcription of genes, causing cell proliferation and survival of the relevant myeloid lineage cells [7]. Over the last 15 years, constitutive activation of JAK2-STAT signaling has been revealed as critical mediator of the MPN pathogenesis [8]. Main mutations including *JAK2* V617F and exon 12, *MPL* and *CALR*, lead to MPN via JAK-STAT constitutional activation [9]. Moreover, negative regulators of this pathway as casitas B-lineage lymphoma proto-oncogene (*CBL*), suppressor of cytokine signaling (*SOCS*) protein, and lymphocyte specific adaptor protein (*LNK*) are altered in MPN [10,11,12]. 

In 2005, the discovery of the *JAK2* V617F mutations by different groups was a major breakthrough into the MPN research. A somatic mutation into exon 14 of *JAK2* gene characterized by a valine to phenylalanine substitution at 617 position results in a conformal change of JH2 pseudo-kinase domain of JAK2. The mutation causes constitutive activation of JAK2-driven signaling pathway in absence of EPOR, MPL, and G-CSFR ligand binding [13,14]. Subsequential downstream activation of intracellular signaling occurs via STAT proteins, mitogen-activated protein kinase (MAPK), and phosphoinositidie-3-kinase (PI3K) [15]. A variety of mouse model demonstrated the *JAK2* role in the MPN pathogenesis in vivo [16]. Through these models (bone marrow transplantation, targeted and transgenic knock-in models), it was clear that *JAK2* V617F mutation is sufficient to drive the disease phenotypes observed in patients. Even the severity of the phenotype is related to the levels of *JAK2* V617F expression, it has not yet been shown how three different diseases could rise from the same mutation. Notably, homozygous mutation and higher mutant allele burden (>50%) have been described associated with an increased risk of thrombosis [17]. The frequency of homozygous mutations varies by 25–30% in PV patients and 2–4% in ET [18].

*JAK2* exon 12 mutations also result in constitutive activation of JAK2 signaling and this occurs in 2–3% of patients with PV [19]. Patients with *JAK2* exon 12 mutations present with most marked erythrocytosis and younger age than V617F mutated, low serum erythropoietin levels, and a distinctive histologic appearance of the bone marrow. The association between different *JAK2* mutations and phenotypical features as been shown in preclinical models as well [20].

In the majority of ET and PMF with *JAK2* mutation wild-type, mutations in *MPL* and *CALR* are detectable. Mutations in *MPL*, located on chromosome 1p34, are present in 1–3% of ET cases and 5% of MF [21]. *MPL* gene contains 12 exon and encodes for TPO-receptor protein. MPL and TPO have a critical role in hematopoietic stem cell self-renewal by increasing DNA-PK-dependent chromosomal integrity and limiting their long-term injury in mouse model [22]. In knockout mice model, MPL also seems to be critical in megakaryocyte development [23]. MPL acts as a regulator of TPO levels providing negative feedback in production of mature platelets. In *MPL* mutated cases, TPO is not cleared, leading to elevated plasma TPO levels, which serves as an unchecked stimulus to drive the observed excessive megakaryocytopoiesis [24]. Several mutations of *MPL* have been identified, but the two most frequent type are W515L and W515K occurring within exon 10 [25]. Variant allele burden of greater than 50% are usually associated with PMF patient and post ET-MF [26].

In 2013, mutations in calreticulin (*CALR*) were identified in two groups of researchers independently by applying whole exon sequencing in JAK2 wild type MPN [27,28]. Calreticulin is an endoplasmic reticulum (ER) chaperone protein, and it is involved in the folding of glycoproteins in the lumen of ER containing a C-terminal ER retention signal with KDEL sequence with negative charges. In *CALR* mutated patients, a shifting of the reading frame leads to a new C-terminal devoid of the KDEL motif, that contain a common new amino acid sequence with positive charges with alteration in calcium homeostasis. Of the more than 50 *CALR* mutations identified, all are located in exon 9 and result in a 1 bp frameshift inducing a novel C-terminal sequence. The most frequent mutations correspond to a 52-bp deletion (p.L367fs*46), also called type 1 in 44% to 53% of patients and a 5-bp insertion (p.K385fs*47), also called type 2 in 32% to 42% of patients. According to these structural changes, the other mutations have been classified as type 1-like and type 2-like [29,30,31,32]. Only patient with ET and PMF harbor *CALR* mutation, suggesting the possible activation of MPL as pathogenic mechanism [33]. Recently, it has been demonstrated that mutant *CALR* induces cytokine independent activation of MPL. The mechanism of interaction and activation of MPL by mutant *CALR* has been described relying on interaction with immature asparagine-linked glycan for engagement with immature MPL in the endoplasmic reticulum. This complex formed between mutant CALR and MPL is then transported to the cell surface, inducing constitutive activation of downstream kinase JAK2 bound to MPL [34]. *CALR* mutation is present in 20–25% of ET and 25–30% of PMF patients, respectively. *CALR* subtypes are associated with peculiar phenotypes and outcomes in MPN. Type 1 mutated patients are more likely to have PMF and with a better survival compared to type 2 mutated PMF. Moreover, type 2 ET patients are associated with higher platelet count [35,36]. Furthermore, *CALR* ET patients compared with JAK2 or *MPL* mutated patients seem to be younger, are more likely to be male and have higher platelet counts, and lower incidence of thrombotic events. Regarding PMF patients, those that harbor *CALR* mutation have younger age, higher platelet count and less anemia compared to those *JAK2* and *MPL* mutated [37]. A recent study suggested unique genetic dependencies from mutant CALR-driven oncogenesis based on N-Glycan biosynthesis pathway. Using pre-clinical model, in vivo the inhibition of N-glycosylation normalized the MPN characteristics in CALR mutated cells [38]. The role of mutated *CALR* in driving the clinical phenotype of MPNs has yet to be fully clarified. CALRdel52 mutations result in increased activation of its acetyltransferase function and upregulation of the transferrin receptor. The latter leads to impairment of iron metabolism inducing a susceptibility to ferroptosis [39].

Despite the discovered of *JAK2*, *CALR*, *MPL* mutations, the insight into the genetic basis of MPN show the presence of approximately 2% of PV and approximately 10% of ET and PMF unmutated for driver mutations. These “triple negative” MPN required a scrupulous diagnostic work-up by excluding reactive causes for a phenotype suggestive of myeloid proliferation. Triple negative ET are typically young female patients. In contrast, tiple negative PMF are associated with poorer prognosis compared to *JAK2*, *MPL* or *CALR* mutated patients [40].

## 3. Additional Mutations in MPN

The previously described driver mutations in *JAK2*, *MPL*, *CALR* cannot fully clarified the heterogeneity of MPNs. With the development of next-generation sequencing, several mutations were identified in more than one-third of MPNs patients [41]. These mutations turned out to be not restricted to MPN and they occur in other myeloid malignancies including myelodysplastic 0syndrome and acute myeloid leukemia. In MPN patients these mutations turn out to have a concrete diagnostic role, in addition to typical bone marrow features [42,43,44]. The most commonly affected genes are those concerning epigenetic regulation, messenger RNA splicing, transcriptional mechanism, signal transduction, and DNA repair mechanism (Table 1) [45]. Discoveries of the somatic mutations in MPN using whole genome analysis implicated a remarkably high number of mutations. This increased availability of genetic sequencing also in the diagnostic setting cleared the genetic heterogeneity of MPN [46,47].

### 3.1. Epigenetic Regulation

DNA methyl transferase 3 (*DNMT3A*) is a member of the family of DNA methyltransferases responsible for the addiction of a methyl group to cytosine in CpG dinucleotides. Several mutations in *DNMT3A* in MPN occur as nonsense/frameshift mutations and missense mutations (including at R882, which is located in the methyltransferase domain), resulting in loss of function [48]. In MPN, *DNMT3A* mutations have been reported in around 10% of patients, with a higher occurrence in PMF [49]. Dnmt3a loss in adult murine models leads to expansion of hematopoietic stem cells (HSC) and cells from progenitor department due to the acquisition of self-renewal ability [50]. Moreover, *DNMT3A* mutations facilitate disease progression in a CRISPR/Cas9 approach, which demonstrated *DNMT3A* loss leads to lethal disease in Jak2V617F-driven MPN mice model by loss of activation of enhancers and aberrant inflammatory signaling [48,51]. The mutation order of *JAK2* V617F and *DNMT3A* mutations is associated with differences in MPN phenotype. Patients are more likely to present with ET compared to PV or PMF when *DNMT3A* mutation are acquired before *JAK2* V617F compared to those who first acquired *JAK2* mutation [52].

Ten-Eleven-Translocation-2 (*TET2*) is an enzyme that converts 5-methylated cytosine to 5-hydroxymethylated cytosine followed by demethylation in DNA [53]. All type of mutations determinate heterozygous or homozygous loss-of-function in its catalytic domain, causing reduced conversion of methylated to hydroxymethylated cytosines. In MPN patients, *TET2* occurs in 7–22% and 19–28% of patients in chronic phase and blast phase, respectively, suggesting that *TET2* mutations could have a role in increase the risk of leukemic transformation [54]. In murine and xenograft models, has been proved a better HSC repopulation in Tet2 mutated HSCs. Moreover, JAK2 mutated/Tet2 mutated HSCT have superior HSC repopulation compared to JAK2 mutated/Tet2 wild type HSCs [55]. The order of acquisition in *JAK2*-mutated MPN with co-occurring mutation in *TET2* impacts the disease phenotype and patients with *JAK2* as first mutations have a higher risk of thrombosis and they have higher probability of presenting with PV than ET with an higher risk of thrombosis [56].

Isocitrate dehydrogenase 1 and 2 (*IDH1/2*) are NADP-dependent enzymes which play a pivotal role in the citric acid cycle and are responsible for catalyzing isocitrate to alpha-ketoglutarate (α-KG) in the cytoplasm (*IDH1*) as well as in the mitochondria (*IDH2*) [57]. Heterozygous missense mutations in the active catalytic site (*IDH1*: R132, *IDH2*: R140 and R172) cause acquisition of the ability to convert α-KG into 2-hydroxyglutatate, interfering with proper TET2 function. *IDH1* and *IDH2* mutations occur at very low frequency in MPNs, but the reported percentage in blast phase MPN are 19–13%. Furthermore, patients who carried *IDH* mutations have worst survival outcome [58]. In murine model, the combined expression of Jak2V617F and mutant *IDH1R132H* or *IDH2R140Q* induces MPN progression, alters stem and progenitor cell function, and impairs differentiation in mice. By combined inhibition of *JAK2* and *IDH2*, stem and progenitor cell compartments were normalized, reducing disease burden better then JAK inhibition alone. These data suggest that combined JAK and IDH inhibition may offer a therapeutic advantage in this high-risk MPNs [59]. The combination of mutations showed impaired differentiation and increased immature progenitors compared to more late stage differentiated progenitors. The *IDH2* mutation has been shown to enhance aberrant splicing of mutant *SRSF2*, leading to genomic instability [60].

Additional Sex Comb Like-1 (*ASXL1*) is involved in epigenetic regulation of gene expression through interaction with PCR2 complex proteins and several other activators and repressors of transcription [61]. Heterozygous nonsense and frameshift mutations in exon 12 lead to loss or gain of function of its PHD domain. ASXL1 associates with the PRC2, and that loss of ASXL1 in vivo collaborates with NRASG12D to promote myeloid leukemogenesis [62]. *ASXL1* mutations are more common in patients with PMF (18–37%) compared to patients with ET and PV (1–11% and 3–12%, respectively) [63]. Asxl1 knockout mice exhibit defects in frequency of differentiation of myeloid progenitors, but did not cause the development of hematological disease phenotype [64]. In a more recent study, *ASXL1* loss in cord blood CD34+ cells reduce erythroid development [65]. Tefferi et al. tried to stratified PMF patients combining the presence of *CALR* and *ASXL1* mutations. Patients with *CALR* mutated/*ASXL1* wild type have the better survival compared to *CALR* wild type/*ASXL1* mutated with a median overall survival of 10.4 years and 2.3 years, respectively [66]. The prognostic role of ASXL1 mutations was recently questioned in MF patients in a recent study, reporting that ASXL1 mutations conferred a worse prognosis only when associated with a high-risk mutation [67]. Furthermore, adult MPN patients with ASXL1 mutations are associated with a significantly higher risk of bleeding, not associated with abnormalities in Von Willebrand factor profile or factor V [68].

Proteins which form part of the polycomb group PRC2 are repressors of transcription through specific post-translational histone modifications. Enhancer of zeste homolog 2 (*EZH2*) is the functional enzymatic component of PRC2. Heterozygous/homozygous loss-of-function mutations disrupt or delete the catalytic SET2 domain, leading to act as a tumor suppressor in MPNs. *EZH2* are found in 1–9% of chronic phase MPN patients. *EZH2* mutated PMF patients had significantly higher leukocyte counts, blast-cell counts, and larger spleen sizes at diagnosis. Leukemia-free survival (LFS) and overall survival (OS) were significantly reduced in *EZH2*-mutated PMF patients [69]. In mice models, the MPN phenotype induced by *JAK2* V617F was accentuated in *JAK2* V617F/*EZH2* (wild type/wild type) mice, resulting in very high platelet and neutrophil counts, more advanced myelofibrosis, and reduced survival [70]. *JAK2* V617F mutation could also interferes with epigenetic processes and recently the role of phosphorylate arginine methyltransferase PRMT5 in myeloproliferative neoplasm (MPN) pathogenesis was investigated. PRMT5 is overexpressed in primary MPN cells, and PRMT5 inhibition potently reduced MPN cell proliferation ex vivo, presenting a potential novel therapeutic target [71].

The High Mobility Group A1 (HMGA1) gene encodes chromatin regulators, and it is overexpressed in MPN patients with progression. In addition, HMGA1 depletion seems to enhance responses to ruxolitinib in murine MF models, and to prolong survival in murine models of JAK2V617F AML, showing HMGA1 as a promising therapeutic target to treat or prevent disease progression [72]. 

### 3.2. Messenger RNA Splicing

The second class of mutations comprises mutations in splicing machinery [73]. Among the mutations involved in RNA splicing *SRSF2*, *U2AF1*, *SF3B1* occur in hot spot regions and nonsense and frameshift mutations are absent, whereas *ZRSR2* harbor only loss-of-function variants. The mechanism by which these somatic mutations lead to splicing abnormalities and different phenotypes still under investigation. 

Serine and arginine rich splicing factor 2 (*SRSF2*) is involved in recognition of exon splicing enhancers. Heterozygous missense mutations and small in-frame deletions in hotspot P95 affect the preferred RNA recognition sequence in RNA exon splicing enhancers by accumulation of R loops, replication stress, and activation of the ATR-Chk1 pathway [74]. The presence of mutated *SRSF2* also affected transcriptional regulation through predominant splicing of RUNX1 to form RUNX1a transcript [75]. In detail, the *RUNX1* gene has several isoforms and the short isoform RUNX1a overexpression has been reported in myeloid disorders [76]. *SRSF2* mutation is found in 3–20% of MPN, with lower frequency in PV and ET compared to PMF and blast phase MPN [77,78,79]. The role of *SRSF2* in MPN pathogenesis and the ability to cooperate with JAK-STAT activating mutations need to be clarified. SRSF2 associated with JAK2 V617F correlates with a reduced leukemia free-survival. In JAK2 V617F transgenic mice, contrary to EZH2 mutation that induce myelofibrotic phenotype, heterozygous Srsf2 delay fibrosis development [80].

U2 small nuclear RNA auxiliary factor 1 (*U2AF1*) is a core part of mRNA splicing machinery and mutations are associated with abnormal splicing of several genes. Most frequent mutations are heterozygous missense mutations around hotspot S34 and Q157.In PMF, *U2AF1* mutations were associated with inferior survival. This datum was confirmed also in MDS patients especially since in these patients the mutation is associated with recognized risk factors, including anemia and thrombocytopenia [81]. *U2AF1* occur in 16% of PMF, and PV and ET patients harboring *U2AF1* mutations have an inferior myelofibrosis-free survival compared to *U2AF1* wild-type patients. The 65% of *U2AF1* mutations affect Q157 and the presence of this mutation is associated with significantly shorter overall survival in MPNs [82]. Besides the canonical function, *U2AF1* has been reported to be able to bind mRNA in the cytoplasm [83].

Splicing factor 3b subunit 1 (*SF3B1*), together with splicing factor 3a and 12S RNA unit, forms the U2 small nuclear ribonucleoproteins complex U2snRPN, which is critical in the early stages of spliceosome assembly. Heterozygous missense mutations are presented in exon 14–16 with hotspot K700E as the most frequent mutations. *SF3B1* mutation is typically presented in patients with MPN/MDS with ring sideroblasts and thrombocytosis (80%) and it occurs only in the 5–10% of MPN patients. *SF3B1* and *JAK2* mutations are commonly observed together in MPN/MDS patients [84]. In mouse model mutant, *JAK2* V617F has been observed to directly phosphorylate components of the splicing machinery in a different way compared to wild type *JAK2*. Moreover, *JAK2* V617F mutant cells are sensitive to JAK inhibitor after activation of splicing enzyme [85]. In combination with *CALR* mutations, *SFR3B1* appears to increase the proliferative advantage of megakaryopoiesis. *CALR* mutant has the ability to bind common major histocompatibility (MHC) class I protein, and the concomitant presence of *SF3B1* leads to higher CALR neoantigen presentation on MHCI, suggesting a potential therapeutic target [86].

Zinc Finger CCCH-Type, RNA Binding Motif and Serine/Arginine Rich 2 (*ZRSR2*) is a gene located on chromosome Xp22.2, which is mutated in about 1–9% of patients with MPN, supporting a role as tumor suppressor.

### 3.3. Transcriptional Regulation

Transcriptional factors are pivotal for regulation of gene expression in MPN patients. Nuclear factor erythroid 2 (*NEF2*), is commonly mutated in MPN patients. The molecular pathogenesis is associated with JMJD1C demethylation and JAK2 phosphorylation [87]. *NEF2* mutations are mostly heterozygous frameshift and they lead to over-expression of wild-type protein functions. 

Runt-related Transcription Factor 1 (*RUNX1*) plays an important role in the regulation of normal hematopoiesis. Somatic mutations of *RUNX1* are frequently found and have been intensively studied in hematological malignancies, such as acute myeloid leukemia (AML), acute lymphoblastic leukemia (ALL), myelodysplastic syndromes (MDS), and chronic myelomonocytic leukemia (CMML). In MPN, *RUNX1* mutations occur in 1–4% of patients. Missense, frameshift, and nonsense mutations cause the loss of function and may act in a dominant-negative fashion over wild-type *RUNX1* [88]. In blast phase MPN, *RUNX1* mutation are demonstrated to be more frequent [89]. *RUNX1* inactivation contributes to AML development through reduced myeloid differentiation. To clarify the molecular mechanisms of evolution, study on ectopic expression of *RUNX1* in CD34+ hematopoietic stem cells from chronic MPN were performed and they showed that RUNX1 transduction resulted in proliferation of immature myeloid cells, enhanced self-renewal capacity, and proliferation of primitive progenitors [90].

### 3.4. Signaling

Beyond the driver mutations with are central driver of JAK-STAT pathway, additional signaling molecules may also be involved in MPN pathogenesis. *RAS* genes encode small GT-Pases with critical roles in cell fate signaling pathways [91]. Different *RAS* genes are mutated in MPN, with *NRAS/KRAS* mutations highly prevalent in these patients. Heterozygous missense substitutions at codon 12, 13, and 61 cause reduced intrinsic GTP hydrolysis and resistance to GAP, driving to activation of growth signaling. Mutations in *NRAS* and *KRAS* in MPNs are likely associated with blast phase and they are present in 7–15% of leukemic transformation. In mice model, the expression of oncogenic K-ras allele caused the development of a MPN phenotype with leukocytosis and normal maturation of myeloid lineage cells, associated with myeloid hyperplasia in bone marrow, and extramedullary hematopoiesis. Furthermore, oncogenic K-ras induces a myeloproliferative disorder but AML, indicating that additional mutations are required [92]. Recent findings showed that the *KRAS* G12D mutation leads to aggressive phenotype of MPN through mediation of Sos1, suggesting the use of Sos1-oncogenic Kras interaction as new therapeutic targets [93]. Protein tyrosine phosphatase, nonreceptor type 11 (*PTPN11*) is a protein encoding a phosphatase which regulates the RAS signaling pathway. Heterozygous missense mutations in the Src-homology 2 (N-SH2) and phosphotyrosine phosphatase (PTP) domains cause increased phosphatase activity. Mutations are found in 6–8% of blast phase MPN [88]. 

Casitas B-cell lymphoma (*CBL*) encodes a RING finger-containing E3 ubiquitin ligases involved in regulation of receptor and nonreceptor tyrosine kinases. Homozygous missense substitutions located in the RING and linker domain reduce E3 ligase activity. Recurrent *CBL* mutations occur in myeloid neoplasms, including 1% to 6% of MPN [94]. Recent data reveal that increased LYN interaction with mutant *CBL* are main factors of enhanced CBL phosphorylation, PI3K regulatory subunit 1 (PIK3R1) recruitment, and (PI3K)/AKT signaling in CBL-mutant cells [95]. 

The lymphocyte adaptor protein (or SH2B adapter protein 3) *LNK* (or *SH2B3*) is an adaptor protein with several domains including SH2 domain can regulate thrombopoietin- MPL-mediated JAK2 activation. Mutations in *LNK* is on example of a negative regulator of JAK/STAT signaling and alterations in exon2 were first described in 6% of V617F JAK2 negative MPN patients [12]. A reduced overall survival has been reported in ET patients harboring *LNK* mutation [96]. In colony-forming unit assays, Lnk, through its SH2 and PH domains, interacts with wild type and mutant *JAK2* and is phosphorylated by constitutively activated *JAK2* V617F. Lnk-deficient murine bone marrow cells are significantly more sensitive to transformation by *JAK2* V617F than wild-type cells. Furthermore, Lnk levels are high in CD34(+) hematopoietic progenitors from MPN and that Lnk expression is induced following JAK2 activation [97]. 

Recently, whole-exome sequencing on CD34+ cells from PMF patients identified a recurrent mutation in complement factor I in 20% of patients, suggesting a role of the complement cascade in the MPN pathogenesis [98]. 

### 3.5. DNA Repair 

Another class of gene important in the molecular pathogenesis of MPN includes genes involved with DNA damage response and cellular stress, which Tumor protein p53 (*TP53*) mutation is the dominant aberration. p53 plays a key role in cell integrity in response to stresses by controlling apoptosis, senescence, DNA repair, or changes in metabolism. Nakatake et al. demonstrated that also the driver mutation *JAK2* V617F upregulate La antigen, increasing MDM2 protein translation and subsequentially altering p53 responses to DNA damage [99]. Mostly missense mutations are found in both allele, which determine a storage of mutant TP53 protein leading to negative effect on wild-type TP53, gain of function and loss of tumor suppression function. *TP53* mutations often indicate blast phase when acquired in MPN [100]. A low burden mutation of *TP53* is reported in chronic MPN but genetic alterations in the tumor suppressor *TP53* are seen in up to 35% of patients upon leukemic transformation [101]. *TP53* mutated MPN patients should be considered a high-risk subgroup of patients that could benefit from a different clinical and therapeutic approaches. 

The serine-threonin protein phosphatase Mg2+/Mn2+ (*PPM1D*) gene negatively regulates TP53 induction in response to DNA damage [102]. This mutation was described in around 2% of MPN, most frequently in patients exposed to chemotherapy [103]. 

Although the three driver mutations *JAK2*, *MPL*, *CALR* are mostly mutually exclusive in MPN, the concomitant presence of two different mutation in *JAK2* gene is reported in few reports. Alternatively, concomitant *MPL* or *CALR* mutations could occur in *JAK2* mutated patients [104]. The clinical implications of these co-occurrences is not clear yet. 

Complex clonal hierarchies have been observed in patients affected by MPN [105]. Acquisition order appears to be determinant in define disease phenotype. Presence of *DNMT3A* and *TET2* mutations confer an advantage to hematopoietic stem/progenitor cells. The epigenetic regulation of transcriptional control affected by mutated *TET2* and *DNMT3A* may allow to HSC to use alternative transcriptional programs and promote self-renewal. In case of *TET2* mutant HSCs, the present of these mutation drives to expansion of the mutant clone in the HSC compartment but without a clear excess production of differentiated erythrocytes and megakaryocytes unit [106]. Furthermore, the mutation order of *JAK2* V617F and *DNMT3A* is associated with differences in MPN phenotype, underlying the importance of the pattern of acquisition of *JAK2* V617F with respect to mutations in epigenetic modifiers in influencing the phenotype of MPN [52]. Additionally, individual with clonal haematopoiesis of indeterminate potential (CHIP) have an enhanced risk of myeloid malignancy, including MPN [107]. In CHIP, *TET2* and *DNMT3A* clonal are present in 25% and 14% smaller than *ASXL1* clonal, respectively. This finding suggests a different ability of *TET2* and *DMT3A* mutation to promote clonal expansion [108].

The role of clonal evolution on the outcome of MPN patients has not been explored yet. Only a large retrospective clinically and biologically real-life study evaluated the acquisition of new additional non-driver mutation during the clinical course, and it demonstrated that the clonal evolution correlate with poor survival in terms of overall survival, progression free survival and secondary MF free survival [109].

## 4. Germline Mutations in MPN

Instead MPN are generally acquired as a result of a somatic mutation with the ability to lead to the clonal expansion of myeloid precursors, several studies have shown familial clustering of MPN with an increased risk of developing MPNs among the relative of patients [110,111]. *JAK2* or *MPL* germline mutations in patients were found in apparently sporadic MPN. Germline genetic factors have been identified to date, including some with rather high frequency in the population but lower penetrance and very rare but highly penetrant mutations clustered in families. Moreover, in MPN germ line mutations in *TERT*, *SH2B3*, *TET2*, *ATM*, *CHEK2*, *PINT*, and *GFI1B* are associated with *JAK2* V617F. These genes impact in several biologic pathways including cellular aging as *TERT*, JAK-STAT signaling as *JAK2, SH2B3*, epigenetic regulation as *TET2*, DNA damage repair and tumor suppressor function as *ATM, CHEK2*, and *PINT*, and erythroid and megakaryocyte development as *GFI1B* [112]. The identification of inherited disease-causing genes might provide new targets for specific therapies.

## 5. Role of Mutations in Leukemic Transformation

Blast phase MPN has a markedly different mutational profile from chronic phase MPN and also from de novo acute myeloid leukemia [113]. Somatic alterations frequently implicated in de novo AML, including *FLT3*, *NPM1*, and *DNMT3A*, are frequently not mutated, instead genes involved in the spliceosome modulator *SRSF2* and in the epigenetic regulation of DNA, including *IDH1/2*, *TET2*, *ASXL1,* and *EZH2* are mutated in bast phase MPN. These findings implied a distinct molecular pathogenesis compared to AML.

Risk factors for leukemic transformation include most aggressive MPN subtype, as MF; other well known risk factors are blast counts above 3–5%, age, anemia, thrombocytopenia, leukocytosis, increasing bone marrow fibrosis, type 1 *CALR*-unmutated status, triple negative status; adverse cytogenetics, and acquisition of ≥2 high-molecular risk mutations (*ASXL1, EZH2, IDH1/2*, *SRSF2*, and *U2AF1Q157*) [114]. Among additional mutations, those affecting *TP53* often coincide with leukemic evolution and synergistic with *JAK2* mutation on leukemogenesis, and they have been associated with a slower, long-term transformation [115]. In contrast, *JAK2* V617F is frequently lost upon leukemic transformation. In case of rapid acute progression, mutations occurring in *RUNX1*, *IDH1/2*, and *U2AF1* have been preferentially described [116]. *ASXL1* mutations have been reported at all phases of disease, suggesting a specific contribution in clonal evolution [54]. *RUNX1* mutations is the main predictor of inferior survival in PMF patients, independent of specific treatment strategies, including hematopoietic stem cell transplant [88].

Recently, elevated dual-specificity phosphatase 6 (DUSP6) protein expression was associated with disease progression in MPN and with high rate of resistance to JAK2 inhibitions. Moreover, the DUSP6-RSK1 axis was proponed as a novel targetable pathway in MPN [117].

Moreover, the presence of any 3 or more somatic mutations has also been shown to predict reduced response to JAK2 inhibitors, suggesting that the presence of multiple mutations might serve influence treatment response and required investigational approaches [118,119].

## 6. Clinical and Molecular-Integrated Prognostic Scores in MPN

Recently, the prognostic relevance of somatic mutations in MPN patients was one of the main research fields in this group of diseases. Adverse molecular variants in ET included *LNK, SF3B1, U2AF1, TP53*, *IDH1*, and *EZH2* impact on overall survival, myelofibrosis-free survival, and leukemia-free survival [120]. A recent study showed the salutary effect of *ASXL1, RUNX1*, and *EZH2* mutations on the risk of arterial thrombosis in ET patients and the prognostic interaction between extreme thrombocytosis and *CALR* mutation in influencing the incidence of arterial events at the time of diagnosis [121]. In PV patients, the presence of *ASXL1, SRSF2*, and *IDH1* seems to be associated with poorer overall survival and leukemia-free survival. [120] PV and ET *JAK2* mutated patients with a persistently high (≥50%) or unsteady *JAK2* V617F load during follow-up have an increased risk of myelofibrotic transformation and a trend for a higher incidence of thrombosis compared to patients with a stable allele burden below 50% [122].

Recently in myelofibrosis mutations played a pivotal role in the development of three new prognostic models in PMF: MIPSS70, MIPSS70+ version 2.0 (MIPSSv2), and GIPSS. These prognostic models add also components that highlighted the independent prognostic contribution of driver or additional mutations. MIPSS70 (mutation-enhanced international prognostic scoring system for transplant-age patients) was based on mutations and clinical variables; MIPSSv2 (the karyotype-enhanced MIPSS70) explored mutation status, karyotype, and clinical variables; GIPSS (the genetically-inspired prognostic scoring system) is based exclusively on mutations and karyotype. (Table 2) This last prognostic score that is exclusively based on genetic (*ASXL1*,* SRSF2*,* U2AF1*, type 1-*CALR*) and cytogenetic markers have a non-inferiority ability in prognostic stratification compared to MIPSS70+ [123,124]. In addition, the specific Myelofibrosis Secondary to PV and ET-Prognostic Model (MYSEC-PM) showed the prognostic role of *CALR* mutation status in secondary MF. Finally, to predict accurately the outcome of transplant candidate MF patients a Myelofibrosis Transplant Scoring System (MTSS) was recently developed. The score also incorporates *CALR* and *MPL* driver mutation and *ASXL1* mutational status [125].

In myelofibrosis, patients presenting with a citopenias involving one or more hematopoietic lineages are defined as MF with myelodepletive phenotype and presented U2AF1 mutations as a distinct molecular marker [126].

## 7. Therapeutic Implications

The integration of molecular knowledges with clinical features is needed to refine disease diagnosis, prognosis, and consequentially to improve rationally derived therapies. (Figure 1) In MF, the discovery of *JAK2* mutation and the observation that also *MPL* and *CALR* mutations induce constitutive activation of JAK-STAT pathway leading to the use of JAK inhibitor ruxolitinib started a new era for the management of these patients. As matter of fact, the development of JAK inhibitors allowed patients to achieve significant advances in control of symptoms and in quality of life improved the quality of life of MF patients, but they are largely insufficient to cure the disease. The main deficiency of ruxolitinib is an absence in clonal selectivity, and efforts are going to generate JAK2-specific inhibitors. In addition, patients relapsed/refractory to ruxolitinib have dismal outcome, in terms of survival and leukemic transformation [127]. JAK inhibitors lack to significantly impact on molecular response and on prevent disease progression [128]. Different JAK inhibitors are currently in indication, such as fedratinib or in advance clinical trials, including pacritinb and momelotinib [129,130,131]. Given the complex pathogenesis, targeted therapies as JAK inhibitors have not been curative. In fact, despite JAK2 inhibitor therapy, other pathway such as MAPK pathway has been shown to remain activated as a compensating process, involving MEK and ERK kinases. Furthermore, targeting MEK/ERK activation pathway seems to increase JAK inhibitor efficacy [132]. Recent data showed that the activation of ERK2 in JAK2V617F mutated MPN enhance PV progression to MF when ERK2 DEF-binding pocket domain function is disable. On the contrary, targeting ERK2 docking D-domain lead to a reduced proliferation of human and murine MPN cells, proving ERK-domain specific role in MPN pathogenesis and supporting development of agents targeting JAK2 and MAPK dependent MPN [133].

However, the role of RON kinase, a member of MET kinase family, is unknown in MPN pathogenesis, the RON phosphorylation was found enhanced in JAK inhibitor persistent cells, suggesting RON inhibitor as a suitable target agent in MPN patients [134].

Recently, Myolas et al. using whole-exome sequencing (WES) at multiple time points showed the acquisition of somatic mutations in MF patients receiving ruxolitinib therapy [135]. Drugs with mutant-specific activity may have limited clinical efficacy due to the complex and dynamic clonal architecture of the MPNs and the role of microenvironment. Similar to management of another disease, as myeloma multiple, the use of combination therapies is already under investigation and the combination therapy will likely be a future topic for treatment, also in upfront setting [136].

Recent molecular acquisitions have allowed the development of new therapeutical strategies [137]. Novel treatments have been developed and are currently in clinical trials for myelofibrosis with targets outside of the JAK-STAT pathway. Multiple pathways are targeted by the next generation of agents for myelofibrosis, including apoptosis (navitoclax, KRT-232, LCL-161, imetelstat), epigenetic modulation (CPI-0610, bomedemstat), and signal transduction pathways (parsaclisib) [138]. New alternative therapeutic targets are now under investigations. A recent paper identified calcium/calmodulin-dependent protein kinase 2 (CAMK2) as a promising therapeutic target in MF patients. In mice model, CAMK2G inhibition ameliorates MF, lessens splenomegaly and leukocytosis, and enhances survival [139].

However, the use of new drugs will soon be required to asses accurate response criteria, and find predictors of response to treatment. Concomitant mutations in MPN such as *IDH1* and *IDH2* could constitute new rationally designed target approaches. Moreover, in MPN with CALR mutated, the exposition of CALR in association with MPL on the cell surface could be used as a therapeutic target [34,140].

## 8. Conclusions

The discovery of the complex molecular landscape provides insight into MPN pathogenesis and reveals novel diagnostic and prognostic markers. Even the driver mutations are central in MPN biology concomitant mutations are common and often associated with more aggressive phenotype. The use of NGS assay in clinical practice is increasing and it aims to predict prognosis and estimate the risk of leukemic transformation. To date, the challenge is translated the complex molecular pathogenesis into effective individualized treatment. The next step will be to answer the unmet needs regarding the understanding of molecular mechanism in patients who lose the response to JAK inhibitor and the identification of new molecular anomalies suitable for target therapy, thus improving molecular-based therapeutic approaches.

## Figures and Tables

**Figure 1 ijms-23-04573-f001:**
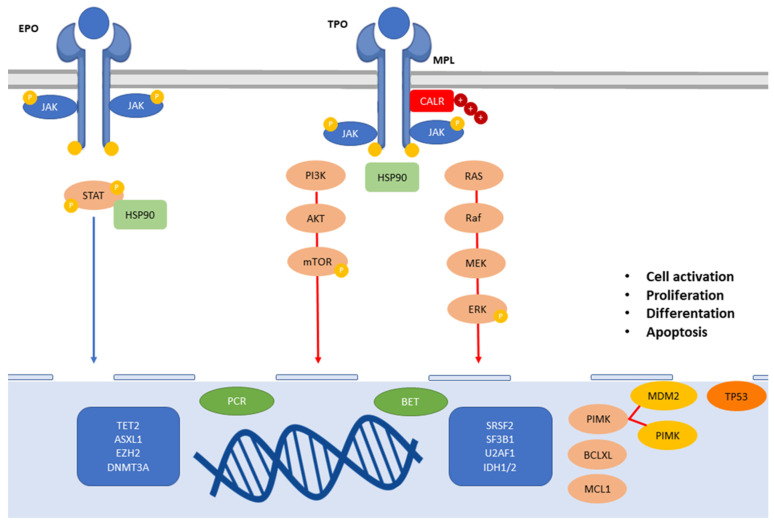
**Signaling pathways involved in the pathogenesis of MPNs**. The complexity of MF disease biology, involving intracelluar proliferative pathways, epigenetic events, HSC maintenance, differentation, and survival mechanisms, leads to the development of different therapeutic approaches.

**Table 1 ijms-23-04573-t001:** Frequencies of additional somatic mutations in MPN.

Class	Mutated Genes	Frequency (%)			
		ET	PV	PMF	Blast Phase
Epigenetic regulation	*DNMT3A*	0–9	0–7	3–15	2–14
	*TET2*	7–16	19–22	10–18	19–28
	*IDH1/IDH2*	1	2	0–6	19–31
	*ASXL1*	1–11	3–12	18–37	17–47
	*EZH2*	1–3	0–3	0–9	13–15
Messenger RNA splicing	*SRSF2*	2	3	8–18	13–22
	*U2AF1*	1	<1	16	5–6
	*SF3B1*	5	3	9–10	4–7
	*ZRSR2*	3	5	10	2
Transcriptional regulation	*NFE2*	<1	2–3	0–3	<1
	*RUNX1*	0–2	0–2	3–4	4–13
Signaling	*NRAS/KRAS*	<1	0–1	3–4	7–15
	*PTPN11*	0–2	<1	0–2	6–8
	*CBL*	0–1	0–2	0–6	4
	*LNK (SH2B3)*	1–3	0–9	0–6	11
DNA repair	*TP53*	2–6	1	1–3	11–36
	*PPM1D*	2	1	1	NA

MPN: myeloproliferative neoplasms; ET: essential thrombocytemia; PV: polycythemia vera; PMF: primary myelofibrosis.

**Table 2 ijms-23-04573-t002:** Prognostic scores in PMF and secondary MF with clinical and molecular feature.

Prognostic Score	Variables (Points)	Risk Categories (Median OS, Years)
MIPSS70	Hemoglobin < 10 g/dL (1) Blasts >2% (1) Constitutional symptoms (1) Leukocytes > 25 × 10*9/L (2) Platelet < 100 × 10*9/L (2) Bone marrow fibrosis ≥ 2 (1) Non type-1 *CALR* (1) HMR = 1 (1) HMR ≥ 2 (2)	0–1: **Low** (27.7) 2–4: **Intermediate** (7.1) 5–12: **High** (2.3)
MIPSS70+	Hemoglobin < 10 g/dL (1) Blasts >2% (1) Constitutional symptoms (1) Non type-1 *CALR* (2) HMR = 1 (1) HMR ≥ 2 (2) Unfavourable karyotype (3)	0–2: **Low** (20.0) 3: **Intermediate** (6.3) 4–6: **High** (3.9) 7–11: **Very high** (1.7)
MIPSS70+ v2.0	Hemoglobin <8–10 g/dL (1) Hemoglobin < 8 g/dL (2) Blasts >2% (1) Constitutional symptoms (2) Non type-1 *CALR* (2) HMR+ *U2AF1Q157* = 1 (2) HMR+ *U2AF1Q157* ≥ 2 (3) HR karyotype (3) VHR karyotype (4)	0: **Very low** (Not reached) 1–2: **Low** (10.3) 3–4: **Intermediate** (7) 5–8: **High** (3.5) 9–14: **Very high** (1.8)
GIPSS	Non type-1 *CALR* (1) *ASXL1* mutated (1) *SRSF2* mutated (1) *U2AF1Q157* (1) HR karyotype (1) VHR karyotype (2)	0: **Low** (26.4) 1: **Intermediate-1** (8) 2: **Intermediate-2** (4.2) 3–6: **High** (2)
MYSEC-PM	Hemoglobin < 11 g/dL (1) Blasts >3% (1) Constitutional symptoms (2) Platelet < 150 × 10*9/L (1) Age at secondary MF (0.15 point/year) *CALR* unmutated (2)	<11: **Low** (Not reached) 11-<14: **Intermediate-1** (9.3) 14-<16: **Intermediate-2** (4.4) ≥ 16: **High** (2)
MTSS	Leukocytes > 25 × 10*9/L (1) Platelet < 150 × 10*9/L (1) Karnofsky performance status <90% (1) Age ≥ 57 years (1) HLA-mismatched unrelated donor (2) Non *CALR/MPL* mutations (2) *ASXL1* mutated (1)	0–2: **Low** (5-years overall survival 83%) 3–4: **Intermediate** (5-years overall survival 64%) 5: **High** (5-years overall survival 37%) 6–9: **Very high** (5-years overall survival 22%)

PMF: primary myelofibrosis, MF: myelofibrosis; MIPSS, Mutation-Enhanced International Prognostic Scoring System; GIPSS, Genetically Inspired Prognostic Scoring System; MYSEC-PM, Myelofibrosis Secondary to PV and ET-Prognostic Model; MTSS, Myelofibrosis Transplant Scoring System. HMR: high molecular risk, include ASXL1, SRSF2, EZH2, IDH1/2; Unfavorable karyotype: any abnormal karyotype other than normal karyotype or sole abnormalities of 20q2, 13q2, +9, chromosome 1 translocation/duplication, -Y or sex chromosome abnormality other than -Y; HR (High risk) karyotype: all the abnormalities that are not VHR and favorable. Favorable karyotype: normal karyotype or sole abnormalities of 20q−, 13q−, +9, chromosome 1 translocation/duplication or sex chromosome abnormality including-Y. Very high risk (VHR) karyotype: single or multiple abnormalities of −7, inv (3), i (17q), 12p−, 11q−, and autosomal trisomies other than +8 or +9.

## Data Availability

Data available in a publicly accessible repository.

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
