# Peer review of "Molecular Pathogenesis of Myeloproliferative Neoplasms: From Molecular Landscape to Therapeutic Implications"

_ijms, 2022, doi:10.3390/ijms23094573_

Round 1

Reviewer 1 Report

The authors have satisfactorily addressed all my concerns.

Author Response

No response needed. 

Reviewer 2 Report

This is a comprehensive review and would seem to have had additional sections added which have contributed to the value of the article.

I would have liked to have seen a figure or two, for example, to illustrate the range of mutation functions

Why is some part of table 1 in bold - it is not explained in the text

Author Response

I added a figure on Signaling pathways involved in the pathogenesis of MPNs. with a short comment (The complexity of MF disease biology, involving intracelluar proliferative pathways, epigenetic events, HSC maintenance, differentation, and survival mechanisms, leads to the development of different therapeutic approaches.)

In table 1, I don't know same data were in bold, in my original table there were not bold items. I think it was a graphic assistent's mistake. 

This manuscript is a resubmission of an earlier submission. The following is a list of the peer review reports and author responses from that submission.

Round 1

Reviewer 1 Report

The Philadelphia-negative myeloproliferative neoplasms are a currently unmet clinical need considered as a disease model to study progression and clinical evolution not only of other leukemias but also of solid tumors. For these reasons, they are the subject of intensive investigation which are frequently and regularly reviewed by giants in the field. A rapid Medline search revealed that only in the last 2 years the reviews published on this subject are 57, with 5 of them dedicated to the mutation landscape of the disease (Dunbar et al, 2020; Szybinski et al, 2021, Palomo et al, 2021, Mullally et al 2020, Hoffman et al, 2021).

Although the mutation landscape reviewed by the current manuscript is a very exciting topic, the manuscript contains little novel information with respect to the reviews, some of which superb, already published in the last 2 years. In fact, only 57 (54%) of the 105 papers summarized by this review were published after 2010, only 8 of which were published after 2020 and therefore less likely to  be included in reviews already published. To be of any value to the field, the authors must: 1) emphasize what are the novelty insights their paper provides to the field, 2) rethink its focus, 3) replace papers published before 2020 with reviews already published and discuss mainly discoveries made after 2020. 

In addition, the review delivers much less than what it promises. The abstract promises a discussion of the relevance of the mutation landscape for therapeutic decision while this subject is briefly covered by one paragraph at the end of the paper. 

Author Response

We decide to go more in detail about the novel therapeutic strategies discussing mainly discoveries made after 2020 in the preclinical field, as you suggested. We want to moving the focus on this paragraph. Moreover, we added same recent review on the references, as you suggest. 

Reviewer 2 Report

The review by Morista et al. on molecular pathogenesis of myeloproliferative neoplasms is an interesting overview of the complex molecular landscape occurring during myeloproliferative neoplasms. They describe the role of driver and other mutations on the pathogenesis and their use in clinic as novel diagnostic and prognostic markers. The review reads very well and suggests novel therapeutic approaches in MPN treatment. I just have few minor points:

  1. Line 113: “also called type 1 in 44% to 53 of patients” the add % after 53.
  2. Line 137: ”Triple negative ET are typically young, female patients.” Take out the “,”after young or replace it by “and”.
  3. Line 179: “in 7%-22% and 19%-28%” please keep a homogeneous way of displaying the % through the manuscript. (e.g in line 83 “varies by 25-30%”).
  4. Line 241: “splicing of RUNX1 to form RUNX1a transcript”. It would nice to introduce here what is
  5. Line 334: “s by controlling apoptosis, apoptosis, senescence,”. The word apoptosis is repeated twice, please take out one.
  6. Line 340: “in MPN(84) Low burden mutation”. A “.” is missing before (84).

Round 2

Reviewer 1 Report

The revisions are minimal (addition of few paragraphs) and do not address the fundamental criticism that is lack of novel insights with respect to recent reviews already published in the field.